# On the Value of Information in Status Update Systems [note 1]

**DOI:** 10.3390/e24040449

**Published:** 2022-03-24

**Authors:** Peng Zou, Suresh Subramaniam

**Affiliations:** Department of Electrical and Computer Engineering, George Washington University, Washington, DC 20052, USA; pzou94@gwu.edu

**Keywords:** age of information, status update system, value of information

## Abstract

The age of information (AoI) is now well established as a metric that measures the freshness of information delivered to a receiver from a source that generates status updates. This paper is motivated by the inherent value of packets arising in many cyber-physical applications (e.g., due to precision of the information content or an alarm message). In contrast to AoI, which considers all packets are of equal importance or value, we consider status update systems with update packets carrying values as well as their generated time stamps. A status update packet has a random initial value at the source and a deterministic deadline after which its value vanishes (called ultimate staleness). In our model, the value of a packet either remains constant until the deadline or decreases in time (even after reception) starting from its generation to the deadline when it vanishes. We consider two metrics for the value of information (VoI) at the receiver: *sum VoI* is the sum of the current values of all packets held by the receiver, whereas *packet VoI* is the value of a packet at the instant it is delivered to the receiver. We investigate various queuing disciplines under potential dependence between value and service time and provide closed form expressions for both average sum VoI and packet VoI at the receiver. Numerical results illustrate the average VoI for different scenarios and relations between average sum VoI and average packet VoI.

## 1. Introduction

In many cyber-physical applications, the need for *real-time* communication of information packets involves not only maintaining information freshness but is also accompanied by the need to preserve the importance or *value* of those packets. Examples of such cases include autonomous cars and general vehicular networks [1,2,3], sensor networks [4,5,6], tactical networks [7] and other systems making decisions in *real-time* [8,9]. In this context, the value of information is another crucial dimension in addition to the notion of timeliness associated with information. In this paper, we address this issue in a queuing system carrying status update packets.

Status update systems with the age of information (AoI) metric measuring end-to-end freshness of packets have received extensive interest recently. Pioneered by the analysis in [10,11] motivated from vehicular status update systems, the AoI metric has been found to be useful in various scenarios such as single server queuing systems [12,13,14], energy harvesting systems [15,16,17,18,19,20], single and multi-hop networks [21,22,23,24,25], cognitive radio [26,27] and vehicular communication networks [28]. The AoI metric provides exclusive meaning to the timing of packets and connects a packet’s usefulness at the receiver with how long the packet spends before its reception. As such, each packet is assumed to be created with the same value starting at generation. The current literature on status update system abstractions is focused mostly on information freshness and does not consider real-time communication of information packets involving a (time-varying) value associated with its content as well as timing, with some attempts in [29,30,31,32,33] being exceptions. In particular, different packets may have different values with respect to the application at the receiver using it. In such scenarios, the AoI metric falls short of capturing all the dimensions of the problem, and a separate *value of information (VoI)* metric has to be introduced.

In this paper, we abstract out the VoI of a status update packet as a time-varying quantity with a random initial value which becomes zero after a deterministic deadline (identical over all packets) inspired by the AoI metric. Packets are assumed to be useless after the deadline, which we term as *ultimate staleness*. We also assume a functional dependence between the initial value of an information packet and its service time to capture the relation between value and data size (e.g., packets carrying higher resolution information are more valuable but larger in size), the growth rate of processes to be monitored (e.g., state estimation in cyber-physical systems) and the content of packets regarding an alarming event. We propose two definitions for VoI. The *sum VoI* is the sum of the current values of all packets held by the receiver, which is reminiscent of throughput. Note that the value of a packet continues to decay after it is received until ultimate staleness. On the other hand, the *packet VoI* is simply the instantaneous value of a packet at the moment it is delivered to the receiver. By comparing the initial value and the packet value, we aim to understand the effect of communication on the lost value.

We note that the use of deadlines has been a topic of research in earlier works in the literature on AoI, motivating us to further explore it in the context of value of information updates. Reference [34] shows how packet deadlines, buffer sizes and packet replacement influence average AoI. Closed-form expressions for average AoI with deadline are derived in [35,36]. Reference [37] studies AoI in a status update system with random packet deadlines and infinite buffer capacity.

Previous works in [29,30,31,32,33] have components related to our view on value of information. For example, references [29,32] consider the quality of information associated with the distortion observed at the receiving end and [38] considers partial updates. Similarly, [31,39] relate the timeliness of observations with the correctness of information. The author of [30] considers age and the value of information with a notion of value taking into account the non-linear costs regarding information updates in various queuing disciplines. The work in [33] evaluates the value of information in addition to age of information in uplink/downlink transmissions in network control systems. The authors of [40] study the performance of VoI and AoI in a first responders’ health monitoring system; their VoI metric is very closely related to our VoI metric originally presented in [41]. In the current paper, we propose a new notion of VoI where a packet’s inherent properties at the time of generation determine its value, in contrast to a value evaluated after processing at the receiver as in previous work. We investigate VoI in M/GI/1/1, M/GI/1/2, M/GI/1/2* and M/GI/1/1* queuing disciplines and provide closed-form expressions for average sum VoI and packet VoI.

The work in this paper is a significantly extended version of our conference paper [41]. In particular, we include the following:We propose and analyze a second VoI metric (average packet VoI) in addition to the average sum VoI analyzed in [41].We add the case of constant value over time until deadline to our analysis on top of the previous work on linear value descent over time until deadline.We analyze the performance of a new queuing scheme, which is M/GI/1/1* in the server. This extended analysis enables us to study the possible use for the value of status update packets in different kinds of systems.We present more numerical results on the two VoI metrics and the four queuing schemes that enable the reader to obtain a clear picture of the various trade-offs involved.

## 2. System Model

We consider a point-to-point communication system with a single transmitter sending status updates from a source to a receiver, as shown in Figure 1. The update packets arrive at the transmitter as a Poisson process with arrival rate λ at instants $t_{i}$. A packet may be discarded in the queuing phase; those that are not discarded enter the server. A packet may also be preempted and discareded while undergoing service; otherwise, it is received by the receiver after system time $T_{i}$ at $t_{i}^{\prime }=t_{i}+T_{i}$. In this paper, we cover M/GI/1/1, M/GI/1/2, M/GI/1/2* and M/GI/1/1* queuing schemes. In M/GI/1/1, there are no buffer and packets arriving in the server-busy state that are discarded. In M/GI/1/2, there is a single data buffer with a first come first serve discipline so that an arriving packet that finds the buffer occupied will be discarded. In M/GI/1/2*, there is a single data buffer but, in this case, an arriving packet will preempt the packet stored in the buffer. In M/GI/1/1*, there are no buffer and packets arriving in the server-busy state that will preempt the current packet in service. For the two no-buffer schemes M/GI/1/1 and M/GI/1/1*, $T_{i}=S_{i}$ where $S_{i}$ is the service time for the *i*th packet, which is independent and identically distributed with $f_{S}\left(s\right)$. For the two schemes with buffer M/GI/1/2 and M/GI/1/2*, $T_{i}=S_{i}+W_{i}$ where $W_{i}$ is the waiting time for the *i*th packet. We derive $T_{i}$ for different schemes in Section 3. We focus on these four queuing systems because previous research has shown that excessive queuing in large buffer systems can adversely impact AoI, and limited-buffer systems with packet management can improve AoI [12,34]. Since the value also potentially becomes worse with time, a similar behavior is expected for VoI.

### 2.1. Value of a Packet

The *i*th update packet has initial value $V_{0,i}$ at the generation instant. This is a random sequence independent over different *i*. $V_{0,i}$ has the identical general distribution $f_{V}\left(v\right)$ with mean value E[V]. This initial value represents the importance of a packet for an application. It could be related to the precision of a measurement, proximity of the sensor to the measured object or it could indicate an alarm event. Each packet has a deterministic lifetime *D* after which it reaches ultimate staleness. Hence, after a fixed time period *D* from packet generation, the packet has no value for the receiver. We use $V_{r,i}$ to denote the instantaneous value of the *i*th update packet when it is delivered to the receiver and $\rho_{i}=\frac{V_{r,i}}{V_{0,i}}$ to denote the fraction of the initial value of the *i*th update packet that is delivered to the receiver.

Motivated by various applications of sensor networking and the value of information in them [1,2,3,4,5,6], in our model, we assume that packet *i*’s value can decrease from its time of generation at $t_{i}$ until it hits the deadline at $t_{i}+D$. The value $V_{i}\left(\tau \right)=h_{i}\left(V_{0,i},\tau \right)$ for the *i*th packet decreases with $\tau =t-t_{i}$, representing the time passed after generation at the transmitter. This value keeps on decreasing (even after a packet is received) until it becomes zero. We have $h_{i}\left(V_{0,i},0\right)=V_{0,i}$ and $h_{i}\left(V_{0,i},D\right)=0$. In this paper, we consider two different *descend functions*
h(.) for the value: (i) constant value and (ii) linear descend. The former models the case where the packet’s value does not change with time as long as it is delivered by the deadline, while the latter models the case where a packet that is delivered earlier has a higher value. In the constant value case, we have the following.
(1)$$\begin{array}{c} V_{i}\left(\tau \right)=h_{i}\left(V_{0,i},\tau \right)=\left\{\begin{array}{cc} V_{0,i}\;\; & \left(\tau <D\right)\\ 0\;\; & (\tau >D). \end{array}\right. \end{array}$$

In the linear case, since $h_{i}\left(V_{0,i},0\right)=V_{0,i}$ and $h_{i}\left(V_{0,i},D\right)=0$, we have a linear descend function.
(2)$$\begin{array}{c} V_{i}\left(\tau \right)=h_{i}\left(V_{0,i},\tau \right)=\left\{\begin{array}{cc} -\frac{V_{0,i}}{D}\tau +V_{0,i}\;\; & \left(\tau <D\right)\\ 0\;\; & (\tau >D). \end{array}\right. \end{array}$$

Then we have the following:(3)$$\begin{array}{c} V_{r,i}=h_{i}\left(V_{0,i},T_{i}\right), \end{array}$$
(4)$$\begin{array}{c} \rho_{i}=\frac{h_{i}\left(V_{0,i},T_{i}\right)}{V_{0,i}}, \end{array}$$
for packets that are deliverd to the receiver. We set $V_{r,i}=0$, $\rho_{i}=0$, for packets that are not delivered to the receiver.

### 2.2. Value-Dependent Service Times

We consider two possibilities for a packet’s service time. In one model, the service times are independent of the initial value of a packet. In another model, the service time of a packet depends on the initial value of the packet through a non-decreasing function *g*.
(5)$$\begin{array}{c} S_{i}=g\left(V_{0,i}\right). \end{array}$$

In this case, the distribution function of $S_{i}$ is $f_{S}\left(s\right)=f_{V}\left(g^{-1}\left(s\right)\right)\frac{dg^{-1}\left(s\right)}{ds}$ where $g^{-1}\left(.\right)$ is the inverse function of g(.), and the mean service time is E[S]=E[g(V)]. Corresponding to the general distribution, we have the moment generating function (MGF) evaluated at −γ for γ≥0:
$$M_{S}\left(\gamma \right)≜E\left[e^{-\gamma S}\right].$$
This monotonic relation reflects the fact that a larger packet takes longer time to transmit and its reception yields more value. This relation causes an interesting tradeoff between value and age as a larger value is obtained at the receiver by paying a longer service time.

In this paper, we consider two definitions for VoI. The first one is $\Upsilon_{\mathrm{sum}}$, which denotes the sum VoI, i.e., the sum of the current values of all packets received by the receiver (cf. [4,5,6] where the additive nature of VoI is discussed in various wireless sensor networks). Hence, $\Upsilon_{\mathrm{sum}}\left(t\right)$ is as follows:(6)$$\begin{array}{c} \Upsilon_{\mathrm{sum}}\left(t\right)=\sum_{j=1}^{i_{t}}V_{j}\left(t\right) \end{array}$$
where $i_{t}=\max\left\{i:t_{i}^{\prime }\leq t\right\}$. The time average of $\Upsilon_{\mathrm{sum}}\left(t\right)$ is the following.
(7)$$\begin{array}{c} E\left[\Upsilon_{\mathrm{sum}}\right]=\lim_{T\to \infty }\frac{1}{T}\int_{t=0}^{T}\Upsilon_{\mathrm{sum}}\left(t\right). \end{array}$$

Another definition is $\Upsilon_{\mathrm{packet}}$, which measures the instantaneous value of a packet at the moment it is delivered to the receiver (if it is delivered). Packets that are dropped are assumed to have zero value. The average packet VoI is then defined as follows.
(8)$$\begin{array}{c} E\left[\Upsilon_{\mathrm{packet}}\right]=E\left[V_{r,i}\right]. \end{array}$$

$E[\rho_{i}]$ is the expected fraction of the initial value that is delivered to the receiver, which illustrates the amount of value received by the receiver compared to the generated initial value at the source. We reiterate that $E[V_{r,i}]$ and $E[\rho_{i}]$ are expectations over *all* packets; dropped packets contribute zero received value.

We illustrate the evolution of value with an example. In Figure 2 and Figure 3, the evolution of value for specific packets generated over time is shown in an M/GI/1/1 system with constant value and linearly descending values, respectively. We use $X_{i}$ to denote the inter-arrival period between two packets i−1 and *i*. Therefore, $X_{i}$ is an exponentially distributed random variable with rate parameter λ. Packet 1 finds the server idle and begins service at $t_{1}$; service ends at $t_{1}^{\prime }$. Packet 2 arrives between $t_{1}$ and $t_{1}^{\prime }$, and it is discarded. The service of packet 1 finishes at $t_{1}^{\prime }$ before the deadline of packet 1, $D_{1}=t_{1}+D$. The value of packet 1 at $t_{1}^{\prime }$, when received by the receiver, is non-zero, and it becomes zero at $D_{1}$. Packets 3, 4 and 5 arrive to the system during the idle period, and they are received at $t_{3}^{\prime }$, $t_{4}^{\prime }$ and $t_{5}^{\prime }$. Note that when packet 4 is received, packet 3 has a non-zero value; thus, the sum VoI, which is shown with a solid red line, is the sum of the values of these packets.

We define areas $Q_{i}$ under the rectangular regions of the curve shown in Figure 2 or the triangular regions of the curve shown in Figure 3, and we set $Q_{i}=0$ for packets discarded in the queuing phase. Then, the expected sum VoI at the receiver is as follows:(9)$$\begin{array}{c} E\left[\Upsilon_{\mathrm{sum}}\right]=\lambda E\left[Q_{i}\right], \end{array}$$
where λ is the arrival rate of packets at the transmitter.

## 3. Evaluating Value of Information

In this section, we derive closed-form expressions for $E[V_{r,i}]$, $E[Q_{i}]$ and E[ρ] for the various queuing systems. $E[\Upsilon_{\mathrm{packet}}]$ and $E[\Upsilon_{\mathrm{sum}}]$ can then be obtained by using Equations (Equation 8) and (Equation 9).

### 3.1. Average VoI for M/GI/1/1

In the M/GI/1/1 queueing system, there is a single server and no buffer. Packets that arrive in the idle period are taken to service immediately and those arriving in busy period are dropped. In view of the renewal structure, we have the following stationary probabilities for each state:(10)$$p_{I}=\frac{1}{\lambda T_{\mathrm{cycle}}},\;p_{B}=\frac{E[S]}{T_{\mathrm{cycle}}},$$
where $T_{\mathrm{cycle}}=\frac{1}{\lambda }+E\left[S\right]$ is the expected length of one renewal cycle; and *I* and *B* indicate the idle and busy states. In the M/GI/1/1 system, packets are delivered to the receiver if they arrive when the server is idle. Recall that if the total time spent by the packet before reaching the receiver is larger than *D*, its value vanishes. Since a packet that is taken to service spends service time $S_{i}$ in the queue before reaching the receiver, the packet’s value vanishes if $S_{i}$ is larger than *D*. Hence, we just need to consider condition $S_{i}<D$ and *i* arriving in idle states. Based on the two time-dependent functions for the value shown in (Equation 1) and (Equation 2) and the relationship shown in (Equation 3)–(Equation 5), we have the following:(11)$$\begin{array}{c} E\left[V_{r,i}\right]=p_{I}\int_{0}^{\tilde{V}}h_{i}\left(v,g\left(v\right)\right)f_{V}\left(v\right)dv, \end{array}$$
(12)$$\begin{array}{c} E\left[\rho_{i}\right]=p_{I}\int_{0}^{\tilde{V}}\frac{h_{i}\left(v,g\left(v\right)\right)}{v}f_{V}\left(v\right)dv, \end{array}$$
(13)$$\begin{array}{c} E\left[Q_{i}\right]=p_{I}\int_{0}^{\tilde{V}}\int_{g(v)}^{D}h_{i}\left(v,\tau \right)f_{V}\left(v\right)d\tau dv, \end{array}$$
where $\tilde{V}=g^{-1}\left(D\right)$ denotes the corresponding initial value when the related service time is equal to the deadline.

### 3.2. Average VoI for M/GI/1/2

In the M/GI/1/2 queueing system, there is a single buffer. The server is in either idle or busy states. Packets that arrive in the idle period are served immediately; those that arrive in the busy period are stored in the buffer if there is no other packet in it and they are discarded otherwise. In view of the renewal structure, we have the following stationary probabilities for each state of the server:(14)$$\begin{array}{c} p_{I}=\frac{1}{\lambda T_{\mathrm{cycle}}},\;p_{B}=\frac{E[S]}{T_{\mathrm{cycle}}M_{S}\left(\lambda \right)}, \end{array}$$
where we use $M_{S}\left(\lambda \right)$ to denote the moment generating function of the service distribution evaluated at −λ:(15)$$\begin{array}{c} M_{S}\left(\lambda \right)=E\left[e^{-\lambda S}\right], \end{array}$$
where $T_{\mathrm{cycle}}=\frac{1}{\lambda }+\frac{E[S]}{M_{S}\left(\lambda \right)}$ is the expected length of one renewal cycle. Next, we evaluate $E[V_{r,i}]$ and $E[Q_{i}|\left(s\right)]$ for $s\in S_{M/GI/1/2}=\left\{I,B\right\}$ and conditioning is on the server state observed by packet *i*. Due to the PASTA property, $Pr\left[P_{i}=\left(s\right)\right]=p_{s}$, where $p_{s}$, $s\in S_{M/GI/1/2}$ are as in (Equation 14).

#### 3.2.1. Idle State Analysis

As a packet arriving in the idle state is served immediately, we have the following.
(16)$$\begin{array}{c} E\left[V_{r,i}|I\right]=\int_{0}^{\tilde{V}}h_{i}\left(v,g\left(v\right)\right)f_{V}\left(v\right)dv, \end{array}$$
(17)$$\begin{array}{c} E\left[\rho_{i}|I\right]=\int_{0}^{\tilde{V}}\frac{h_{i}\left(v,g\left(v\right)\right)}{v}f_{V}\left(v\right)dv, \end{array}$$
(18)$$\begin{array}{c} E\left[Q_{i}|I\right]=\int_{0}^{\tilde{V}}\int_{g(v)}^{D}h_{i}\left(v,\tau \right)f_{V}\left(v\right)d\tau dv. \end{array}$$

#### 3.2.2. Busy State Analysis

Since only the first packet that arrives during the busy period is served and the others are discarded, we introduce a lemma for the probability that an arriving packet is the first one that arrives in the busy state. To do so, we first define states $B_{1}$ and $B_{2}$ as the busy states of the server with zero and one packet waiting in the queue, respectively. The renewal cycle is as follows. After the idle period, an arrival happens and the system turns to $B_{1}$ state. Now, a time duration of service *S* starts and if during the service period another arrival occurs, the system turns to $B_{2}$ state. This back-and-forth between $B_{1}$ and $B_{2}$ states continues until no packet arrives in one service time. We provide an example in Figure 4 for the three states in the M/GI/1/2 scheme. At time $t_{0}$, packet 1 arrives and finds the system idle. Packet 2 finds the system in $B_{1}$ state at $t_{1}$ and is stored in the buffer. Packet 3 finds the system in $B_{2}$ state at $t_{2}$ and is dropped.

This renewal structure yields the following result.

**Lemma** **1.**
*In the M/GI/1/2 scheme, the waiting time of a packet in the buffer conditioned on its arrival in $B_{1}$ state is as follows*

$$\begin{array}{cc} E[W_{B_{2}}] & =E[S-X|X<S]Pr[X<S]\\ \; & =E\left[S\right]+\frac{1}{\lambda }M_{S}\left(\lambda \right)-\frac{1}{\lambda }. \end{array}$$


*The stationary probability of $B_{2}$ state is as follows:*

$$\begin{array}{c} p_{B_{2}}=p_{B}\frac{E[W_{B_{2}}]}{E[S]}=p_{B}\left(1+\frac{M_{S}\left(\lambda \right)-1}{\lambda E[S]}\right), \end{array}$$

*and the probability of $B_{1}$ state is $p_{B_{1}}=p_{B}-p_{B_{2}}$.*


Then, we have $E\left[Q_{i}|B\right]=E\left[Q_{i}|B_{1}\right]$ and we provide the probability distribution function for the conditional residual service time $W^{{}^{\prime }}$ under the condition that the packet arrives in the $B_{1}$ state:$$\begin{array}{cc} P[W^{{}^{\prime }}>w] & =P[S-X>w|X<S]\\ & =\frac{\int_{w}^{\infty }\int_{0}^{s-w}f_{S}\left(s\right)f_{X}\left(x\right)dxds}{P[X<S]}\\ \; & =\frac{\int_{w}^{\infty }f_{S}\left(s\right)\left(1-e^{-\lambda (s-w)}\right)ds}{1-M_{S}\left(\lambda \right)}, \end{array}$$
and we have the following.
(19)$$\begin{array}{c} f_{W^{{}^{\prime }}}\left(w\right)=\frac{d(1-P\left[W^{{}^{\prime }}>w\right])}{dw}. \end{array}$$
Then, we have the following.
(20)$$\begin{array}{c} E\left[V_{r,i}|B_{1}\right]=\int_{0}^{\tilde{V}}\int_{0}^{D-g(v)}h_{i}\left(v,g\left(v\right)+w\right)f_{W^{{}^{\prime }}}\left(w\right)f_{V}\left(v\right)dwdv, \end{array}$$
(21)$$\begin{array}{c} E\left[\rho_{i}|B_{1}\right]=\int_{0}^{\tilde{V}}\int_{0}^{D-g(v)}\frac{h_{i}\left(v,g\left(v\right)+w\right)}{v}f_{W^{{}^{\prime }}}\left(w\right)f_{V}\left(v\right)dwdv, \end{array}$$
(22)$$\begin{array}{c} E\left[Q_{i}|B_{1}\right]=\int_{0}^{\tilde{V}}\int_{0}^{D-g(v)}\int_{g(v)+w}^{D}h_{i}\left(v,\tau \right)f_{W^{{}^{\prime }}}\left(w\right)f_{V}\left(v\right)d\tau dwdv. \end{array}$$

Therefore, we have $E\left[V_{r,i}\right]=E\left[V_{r,i}|I\right]p_{I}+E\left[V_{r,i}|B_{1}\right]p_{B_{1}}$, $E\left[\rho_{i}\right]=E\left[\rho_{i}|I\right]p_{I}+E\left[\rho_{i}|B_{1}\right]p_{B_{1}}$ and $E\left[Q_{i}\right]=E\left[Q_{i}|I\right]p_{I}+E\left[Q_{i}|B_{1}\right]p_{B_{1}}$.

### 3.3. Average VoI for M/GI/1/2*

The M/GI/1/2* queueing system is the same as M/GI/1/2 except that we use a last-come first-serve order with packet discarding in the buffer. The latest packet arriving in a busy period takes the place of the old packet in the buffer. Therefore, we have the same stationary probabilities for each state as the M/GI/1/2 system in (Equation 14). Additionally, the expressions for $E[V_{r,i}|I]$, $E[\rho_{i}|I]$ and $E[Q_{i}|I]$ are the same as in (Equation 16)–(Equation 18) separately. We now derive expressions for $E[Q_{i}|B]$ and $E[V_{r,i}|B]$.

#### Busy State Analysis

If the *i*th packet arrives to the server during the busy period, it will be transmitted to the receiver conditioned on event $\left\{X_{i}>W_{i-1}\right\}$, which means the next packet arrives for the server after the current service finishes. *W* is the general residual service time for all packets arriving in the busy state, and we have the following: $f_{W}\left(w\right)=\frac{P[S>w]}{E[S]}$. Then, the following is the case.
(23)$$\begin{array}{cc} E\left[V_{r,i}|B\right]=\int_{0}^{\tilde{V}}\int_{0}^{D-g(v)}\int_{w}^{\infty } & h_{i}\left(v,g\left(v\right)+w\right)f_{X}\left(x\right)\\ & f_{W}\left(w\right)f_{V}\left(v\right)dxdwdv, \end{array}$$
(24)$$\begin{array}{cc} E\left[\rho_{i}|B\right]=\int_{0}^{\tilde{V}}\int_{0}^{D-g(v)}\int_{w}^{\infty } & \frac{h_{i}\left(v,g\left(v\right)+w\right)}{v}f_{X}\left(x\right)\\ & f_{W}\left(w\right)f_{V}\left(v\right)dxdwdv, \end{array}$$
(25)$$\begin{array}{cc} E\left[Q_{i}|B\right]=\int_{0}^{\tilde{V}}\int_{0}^{D-g(v)}\int_{w}^{\infty }\int_{g(v)+w}^{D} & h_{i}\left(v,\tau \right)f_{X}\left(x\right)\\ & f_{W}\left(w\right)f_{V}\left(v\right)d\tau dxdwdv. \end{array}$$

Therefore, we have $E\left[V_{r,i}\right]=E\left[V_{r,i}|I\right]p_{I}+E\left[V_{r,i}|B_{1}\right]p_{B_{1}}$, $E\left[\rho_{i}\right]=E\left[\rho_{i}|I\right]p_{I}+E\left[\rho_{i}|B_{1}\right]p_{B_{1}}$ and $E\left[Q_{i}\right]=E\left[Q_{i}|I\right]p_{I}+E\left[Q_{i}|B_{1}\right]p_{B_{1}}$.

### 3.4. Average VoI for M/GI/1/1*

In the M/GI/1/1* queueing system, there is no buffer and a new packet that arrives during busy state will preempt the current packet in service. Since the arrival process is a Poisson with rate λ, $p_{e}$, the probability that a packet is delivered to the receiver is given by the following:(26)$$\begin{array}{c} p_{e}=P\left[S_{i}<X_{i+1}\right]=M_{S}\left(\lambda \right), \end{array}$$
which means, in preemption scheme, only the packet that has a service time less than the upcoming inter-arrival period is delivered to the receiver. We use relation $f_{G|G<F}\left(t\right)=f_{G}\left(t\right)\frac{P(F>t)}{P(G<F)}$ from [13] where *G* and *F* are arbitrary random variables. Since $P\left(G<F\right)=M_{G}\left(\lambda \right)$ and $P\left(F>t\right)=e^{-t\lambda }$, we have the probability density function for conditional service time.
(27)$$\begin{array}{c} f_{S|S<X}\left(s\right)=f_{S}\left(s\right)\frac{e^{-s\lambda }}{M_{S}\left(\lambda \right)}. \end{array}$$

We use $S^{\prime }$ to denote the conditional service time *S*; therefore, we have $f_{S^{\prime }}\left(s\right)=f_{S|S<X}\left(s\right)$. In this case, we rewrite Equation (Equation 1) as follows:(28)$$\begin{array}{c} h_{i}\left(g^{-1}\left(s\right),\tau \right)=\left\{\begin{array}{cc} g^{-1}\left(S_{i}^{\prime }\right)\;\; & \left(\tau <D\right)\\ 0\;\; & \left(\tau >D\right) \end{array}\right. \end{array}$$
and Equation (Equation 2) as the following.
(29)$$\begin{array}{c} h_{i}\left(g^{-1}\left(s\right),\tau \right)=\left\{\begin{array}{cc} -\frac{g^{-1}\left(S_{i}^{\prime }\right)}{D}\tau +g^{-1}\left(S_{i}^{\prime }\right)\;\; & \left(\tau <D\right)\\ 0\;\; & \left(\tau >D\right) \end{array}\right. \end{array}$$

Then, we have the following.
(30)$$\begin{array}{c} E\left[V_{r,i}\right]=p_{e}\int_{0}^{D}h_{i}\left(g^{-1}\left(s\right),s\right)f_{S^{\prime }}\left(s\right)ds, \end{array}$$
(31)$$\begin{array}{c} E\left[\rho_{i}\right]=p_{e}\int_{0}^{D}\frac{h_{i}\left(g^{-1}\left(s\right),s\right)}{g^{-1}\left(s\right)}f_{S^{\prime }}\left(s\right)ds, \end{array}$$
(32)$$\begin{array}{c} E\left[Q_{i}\right]=p_{e}\int_{0}^{D}\int_{s}^{D}h_{i}\left(g^{-1}\left(s\right),\tau \right)f_{S^{\prime }}\left(s\right)d\tau ds, \end{array}$$

## 4. Numerical Results

In this section, we provide numerical results for average VoI for various cases. We also perform packet-based queue simulations offline for $10^{6}$ packets as verification of the analytical results. An example of our simulation results is shown in Figure 5. We use g(V)=V as the relation between service time and value to model the case where the value is directly proportional to the packet size. Results are presented for three different distributions for the initial value of packets.

### 4.1. Uniformly Distributed Initial Value

First, we assume that the initial value of each packet is uniformly distributed between $V_{\min}$ and $V_{\max}$ and the value follows the linear descend function. In Appendix A, we provide closed-form expressions for $E[\Upsilon_{\mathrm{sum}}]$ and $E[\Upsilon_{\mathrm{packet}}]$ in various systems with linearly descending value.

We show a comparison of average $\Upsilon_{\mathrm{sum}}$ and average $\Upsilon_{\mathrm{packet}}$ in Figure 5. In Figure 5a, we show average $\Upsilon_{\mathrm{sum}}$ versus arrival rate λ for the four queuing schemes. We observe that M/GI/1/1 and M/GI/1/2* perform better than M/GI/1/1* as λ increases. In particular, due to the linear relation between time and value, keeping a packet in the buffer to keep the server busy turns out to yield smaller value at the receiver with respect to keeping none and serving only the freshest packets. For M/GI/1/1* and M/GI/1/2, on the other hand, there is an optimal value of λ after which average $\Upsilon_{\mathrm{sum}}$ drops. For M/GI/1/2, it is due to undesired increases in waiting times in the data buffer while for M/GI/1/1*, it is due to undesired decrease in the number of delivered packets.

In Figure 5b, we show $\Upsilon_{\mathrm{packet}}$ versus arrival rate λ for the four queuing schemes. Again, we observe that M/GI/1/1 performs better than the other three.We observe that as λ increases, $E[\Upsilon_{\mathrm{packet}}]$ decreases in all four queuing schemes due to the fact that most of the generated packets are discarded in the queuing phase and have zero value for the receiver.

In Figure 6, we show E[ρ], which denotes the average ratio of the received value compared to the generated values over all the generated packets. We observe that as λ increases, E[ρ] decreases in all four queuing schemes, which matches the result for $E[\Upsilon_{\mathrm{packet}}]$. However, interestingly, M/GI/1/1* scheme performs best for E[ρ]. This is because as λ increases, even though there will be more packets dropped, the packets delivered to the receiver have smaller service times, which increases the ratio of the delivered value to the initial value.

Next, we consder the case when the service times are independent of the initial values and are exponentially distributed with service rate μ. In Figure 7, we show the average $\Upsilon_{\mathrm{sum}}$ versus arrival rate λ for the four queuing schemes. We observe that M/GI/1/1* performs better than the other three.This is because the service time is independent of the initial value, and large-valued packets may have small service times. In particular, due to the linear relation between time and value, keeping a packet in the buffer to keep the server busy turns out to yield smaller values at the receiver compared to keeping none and serving only the freshest packets.

Finally, in Figure 8, we show the average $\Upsilon_{\mathrm{sum}}$ versus service rate μ for the four queuing schemes when the service times are independent of the initial values and are exponentially distributed. We observe that M/GI/1/2 and M/G/1/2* perform better than M/GI/1/1* as μ increases. This is because, as the average service time deceases, fewer packets will expire, i.e., reach ultimate staleness, during the waiting period in the buffer, and in this case, having a buffer to store the packets turns out to yield larger value at the receiver with respect to dropping the packets in the server.

### 4.2. Exponentially Distributed Initial Value

Next, we consider $f_{V}\left(v\right)=\mu_{v}e^{-\mu_{v}v}$ with constant value. In this case, we have service rate $\mu =\mu_{v}$ due to g(V)=V. We compare average AoI with average sum VoI for the same schemes as both of them are time-average metrics over all the packets. In Appendix B, we provide closed-form expressions for $E[\Upsilon_{\mathrm{sum}}]$ and $E[\Upsilon_{\mathrm{packet}}]$ in various systems for constant values.

In Figure 9a, we plot the average $\Upsilon_{\mathrm{sum}}$ with respect to λ for various schemes. We observe that M/M/1/2* always performs better than the others. This is connected to the fact that when the value of packet is constant over time, all packets received within the deadline contribute their full initial value. Since $\Upsilon_{\mathrm{sum}}$ is the accumulated value of received packet values, the total value is higher if a packet is stored in the buffer instead of dropping it. At the same time, we observe that M/M/1/1* performs the worst in terms of value since the dependence between service time and value causes higher value packets to be preempted in this system, resulting in no contribution to VoI at the receiver.

Next in Figure 9b, we show average $\Upsilon_{\mathrm{sum}}$ for independent initial value and service time under the same marginal distributions. We observe that, with independent service time, the M/M/1/1* scheme becomes the best case while it is the worst case with dependent service time. The other three schemes yield higher values as the adverse relation between initial value and service rate is removed.

Finally, in Figure 10, we show E[ρ] versus deadline *D* for the four queuing schemes. We observe that, as *D* increases, E[ρ] for all queuing schemes increases, but never reaches threshold 1 due to the fact that some packets are discarded in the queuing phase.

### 4.3. Binary Distributed Initial Value

We finally consider binary distributed initial value for two classes of update packets. Class 1 and class 2 packets have $V_{0,i}=V_{1}$ and $V_{0,i}=V_{2}$. Each packet is independently chosen to be in class 1 or 2 with probability *p* and (1−p), respectively. This situation models the case when a packet of one class contains a message about an alarming event yielding high value once received, whereas the other class of packets are assumed to be regular status updates.

In Figure 11, we set $V_{1}=1.33$, $V_{2}=0.4$ and p=0.2. We compare plots showing average $\Upsilon_{\mathrm{sum}}$ versus λ for three different service policies in an M/M/1/1 system. The first policy serves all packets without regard to the value, the second policy involves serving only class 1 packets, and the third policy serves only class 2 packets. Note that if the service time is dependent on the value, class 1 packets will have exponentially distributed service time with mean $E\left[S\right]=E\left[V_{1}\right]$, and similarly, class 2 packets will have exponentially distributed service time with mean $E\left[S\right]=E\left[V_{1}\right]$. If the service time is independent of the value, both class packets will have exponentially distributed service time with μ=1.5. Our numerical results show that when service time is independent of value, always serving the high-value packet will yield the highest average value. On the other hand, in the dependent case when arrival rate becomes large, serving the packet with low value but smaller service time and high probability will benefit the average $\Upsilon_{\mathrm{sum}}$ compared to serving all the packets or serving the high-value packets with larger service time and low probability.

## 5. Conclusions

Age of information (AoI) is a well-known metric that quantifies the freshness of information at a receiver in status update systems. This metric ignores the potential differences in the importance of various update packets. In this paper, we consider the value of information in status update systems wherein packets have various initial values upon generation. We investigate various queuing disciplines with initial-value-dependent packet service times and obtain closed-form expressions for two different VoI metrics. Our numerical results illustrate the trade-off between the two VoI metrics and the contrast between these two metrics. We show average sum VoI and average packet VoI for different scenarios and the fraction of received value comparing to the inital value for different systems.

## Figures and Tables

**Figure 1 entropy-24-00449-f001:**
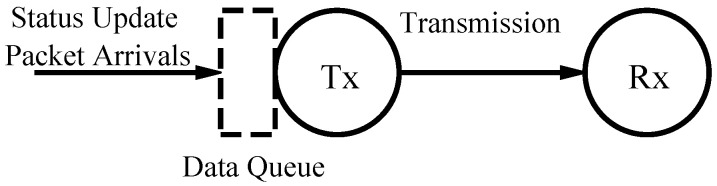
System model with status update packets arriving at a single server transmission queue.

**Figure 2 entropy-24-00449-f002:**
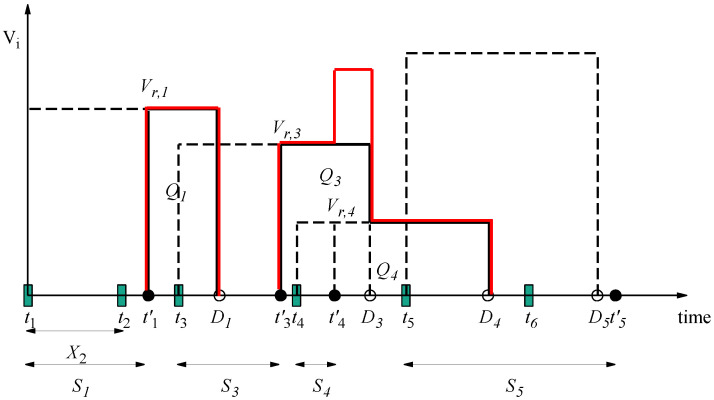
Evolution of value in M/GI/1/1 system when the value remains constant until deadline.

**Figure 3 entropy-24-00449-f003:**
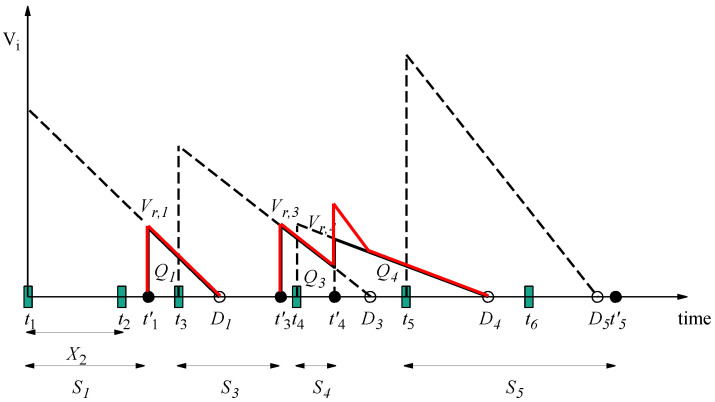
Evolution of thevalue in the M/GI/1/1 system with linearly descending values.

**Figure 4 entropy-24-00449-f004:**
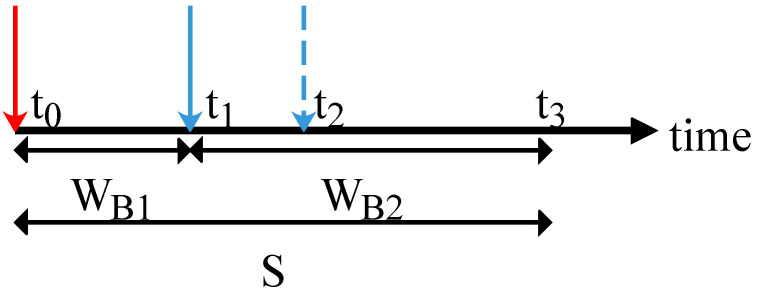
Three states that can be observed by packets in M/GI/1/2 scheme.

**Figure 5 entropy-24-00449-f005:**
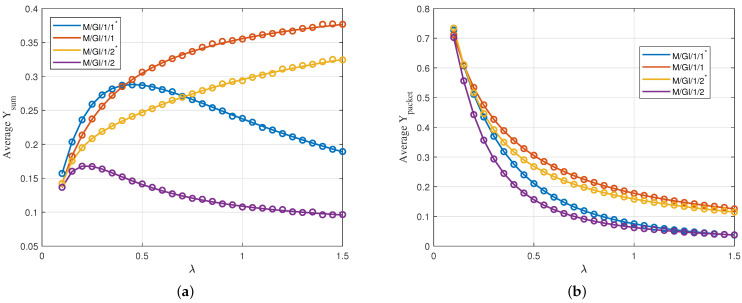
(**a**) Average $\Upsilon_{\mathrm{sum}}$ and (**b**) average $\Upsilon_{\mathrm{packet}}$ for uniformly distributed initial value with linear descend function versus λ; $V_{\min}=0$, $V_{\max}=10$, D=8. Circles are simulation results.

**Figure 6 entropy-24-00449-f006:**
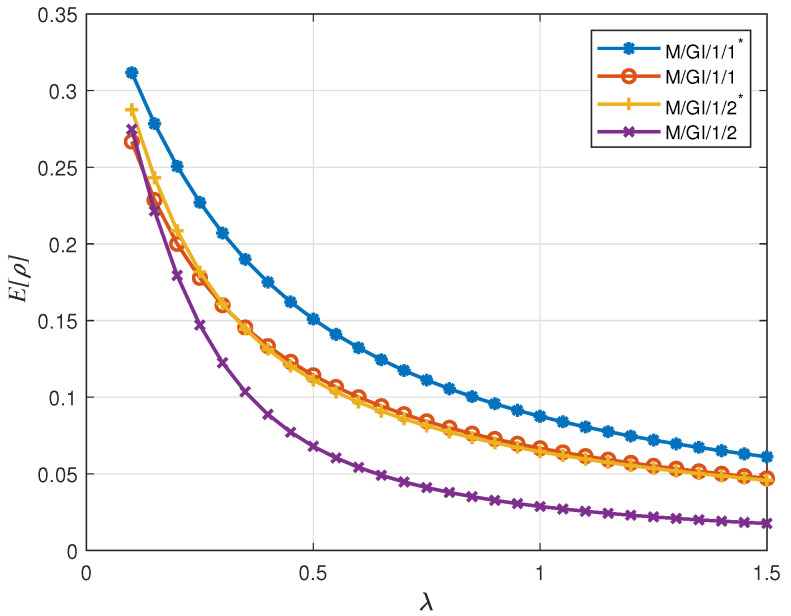
E[ρ] for uniformly distributed initial value with linear descend function versus λ; $V_{\min}=0$, $V_{\max}=10$, D=8.

**Figure 7 entropy-24-00449-f007:**
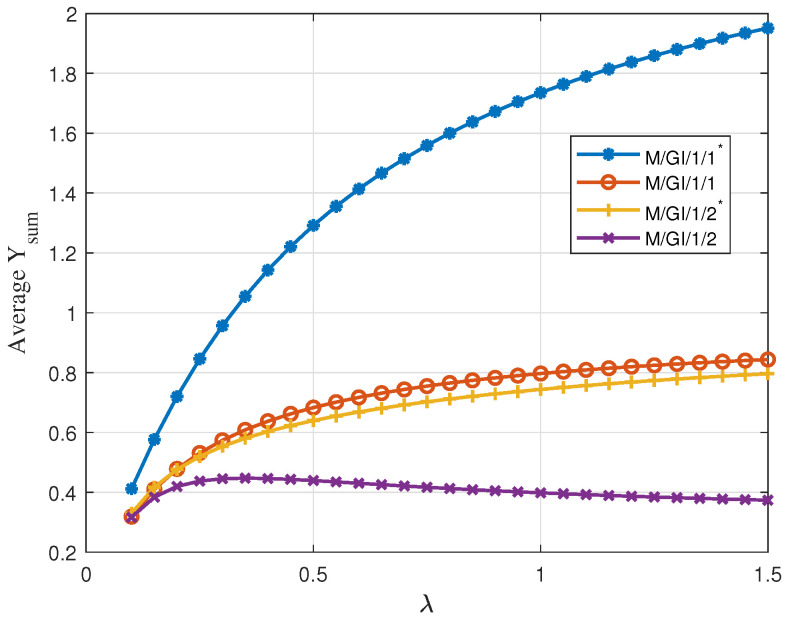
Average $\Upsilon_{\mathrm{sum}}$ for uniformly distributed initial value with linear descend function and exponential independent service time versus λ; $V_{\min}=0$, $V_{\max}=10$, D=8, μ=0.2.

**Figure 8 entropy-24-00449-f008:**
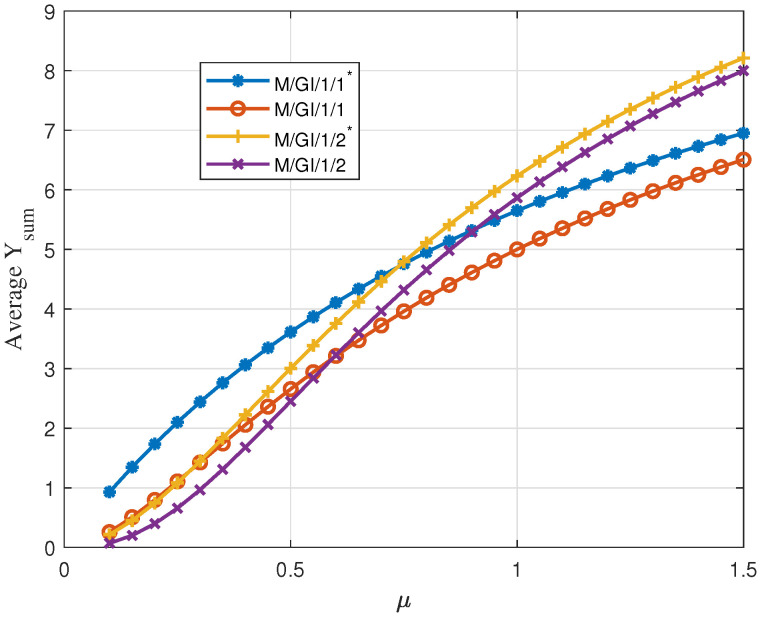
Average $\Upsilon_{\mathrm{sum}}$ for uniformly distributed initial value with linear descend function and exponential independent service time versus μ; $V_{\min}=0$, $V_{\max}=10$, D=8, λ=1.

**Figure 9 entropy-24-00449-f009:**
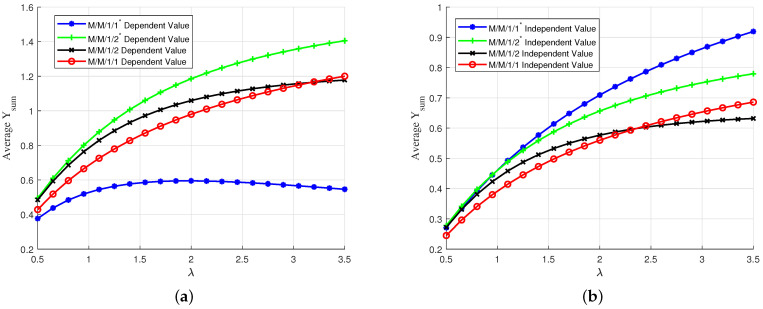
Average $\Upsilon_{\mathrm{sum}}$ for exponentially distributed service time with constant value versus λ for M/M/1/1, M/M/1/2, M/M/1/2* and M/M/1/1* schemes with $\mu_{v}=1.5$ and D=3. (**a**) Dependent Value. (**b**) Independent Value.

**Figure 10 entropy-24-00449-f010:**
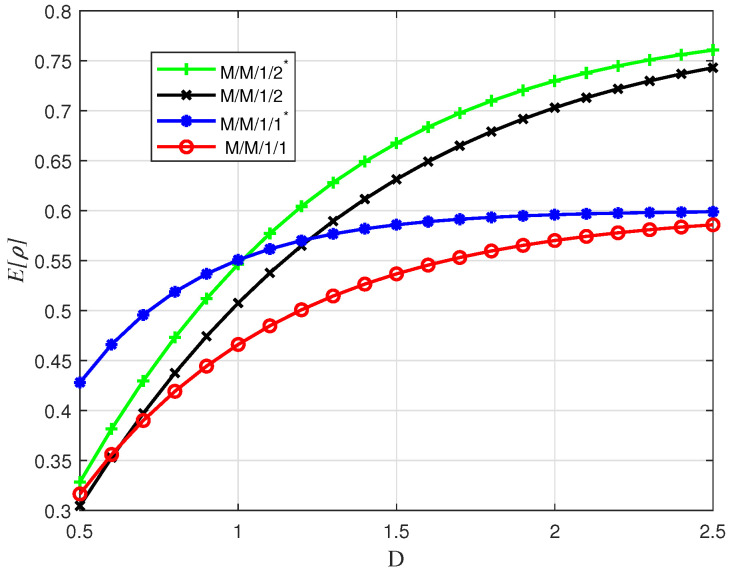
E[ρ] for exponentially distributed service time with constant value versus *D* for λ=1.

**Figure 11 entropy-24-00449-f011:**
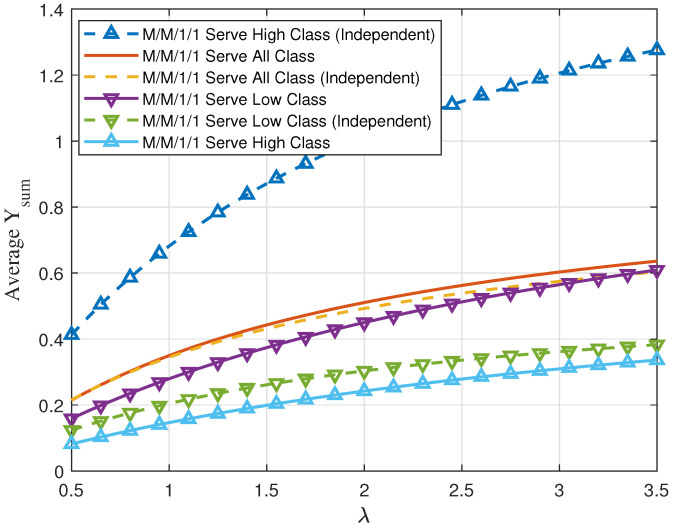
Exponentially distributed service time dependent on or independent of the binary value in M/M/1/1 scheme.

## Data Availability

Not applicable.

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
