# Peer review of "On the Value of Information in Status Update Systems†"

_entropy, 2022, doi:10.3390/e24040449_

Round 1

Reviewer 1 Report

The paper is considering the case of systems where a source generates status updates, and the system has control over the queuing and delivery of the packets to the interested receiver. The authors are considering a system in which the value of information associated with a packet is being considered. The value starts out as being random at the moment it is generated at the source but it decreases in time linearly, reaching the value of zero at a predefined deadline.
The paper is considering different techniques for optimizing the VoI, focusing on two optimization criteria: the VoI of a packet at delivery time and the sum VoI of all the packets held by the source at a given moment in time. Considering a variety of queuing theory models, the authors are deriving closed form expressions for these two metrics. These closed form models are then verified using numerical simulations under different assumptions of the distribution of the initial value (eg. uniform, exponential and binary distributions).
The paper is well written. The arguments are clearly presented.

Author Response

We deeply appreciate the reviewer’s appreciation and hard work. We have fixed some typos to make the revised paper in good shape.

Reviewer 2 Report

This paper discusses the sum VoI and packet VoI under various queuing disciplines and different potential dependence between value and service time. The closed-form expressions of both sum VoI and packet VoI are given, and the inherent relations and comparative insights are investigated. In general, this paper is easy-to-follow and well-written. This paper is recommended to be accepted if the following suggestions were addressed.
1. The authors should further emphasize the contributions and research motivations of this work. In particular, as VoI has been studied since 2013, the novelty of this work seems to be investigating the VoI with packet deadlines; however, the necessity to discuss the packet deadline is not clearly stated in this paper. Please elaborate this.
2. The buffer size is set as 1 or 2 in this paper. However, some recent works have discussed the AoI with larger buffer sizes or even infinite buffer capacity. If possible, could the authors provide detailed analyses of VoI with deadlines under larger buffer sizes?
3. The authors considers two different descend functions h(.) in this paper, one case is the constant value setup, the other case performs in linear descend. However, the physical meaning of such cases are not elaborated in this paper. Please provide detailed explanations. 
4. Also, it would be better to elaborate the detailed applications scenarios of different distributions for the initial value in Section 4.
5. The authors state that they perform packet-based queue simulations offline for 10^6 packets as verification of the analytical results. However, the simulation results are not demonstrated. Please supplement the simulation results in Section 4.
6. There are some typos in this paper, e.g., in line 81 of page 2, “queuing queuing scheme” should be “queuing scheme”; in Figure 5, Figures 7-9 and Figure 11, the labels of the longitudinal axis should be unified as Ysum. Please carefully recheck them.

Author Response

Thank you for your precious comments and advice. Those comments are all valuable and very helpful for revising and improving our paper, as well as the important guiding significance to our researches. We have studied comments carefully and have made correction which we hope meet with approval. Revised portion are marked in blue in the paper. The main corrections in the paper and the responds to the reviewer’s comments are as flowing:

  1. The authors should further emphasize the contributions and research motivations of this work. In particular, as VoI has been studied since 2013, the novelty of this work seems to be investigating the VoI with packet deadlines; however, the necessity to discuss the packet deadline is not clearly stated in this paper. Please elaborate this.

We thank the reviewer for the comment. VoI has been studied since 2013, but how to connect the timeliness with instantaneous value of a packet continues to be a problem of interest for many researchers. As the AoI linearly increases with time, we formulate the VoI metric in a similar way wherein the VoI decreases with time. We note that the use of deadlines has been a topic of research in earlier works in the literature on AoI, motivating us to further explore it in the context of value of information updates where the VoI of a packet will become zero when it passes the deadline.

  1. The buffer size is set as 1 or 2 in this paper. However, some recent works have discussed the AoI with larger buffer sizes or even infinite buffer capacity. If possible, could the authors provide detailed analyses of VoI with deadlines under larger buffer sizes?

We thank the reviewer for the comment. In this paper, we consider these four queueing schemes since limited-buffer systems with packet management can improve AoI and we expect a similar behavior for VoI. The analysis of large-buffer systems requires new methods, and we expect to report them in the future   

  1. The authors considers two different descend functions h(.) in this paper, one case is the constant value setup, the other case performs in linear descend. However, the physical meaning of such cases are not elaborated in this paper. Please provide detailed explanations. 

We thank the reviewer for raising a very good point. In our revised paper, we have added a sentence in Section 2.1 to clarify our motivation for these two functions..

  1. Also, it would be better to elaborate the detailed applications scenarios of different distributions for the initial value in Section 4.

This is a good point. The binary initial value denotes a two-state system when the system is in normal state or abnormal state. For the uniform and exponential distribution value, those are widely studied queue model and we connect the value to service time to model the case where the value is directly proportional to the packet size. 

  1. The authors state that they perform packet-based queue simulations offline for 10^6 packets as verification of the analytical results. However, the simulation results are not demonstrated. Please supplement the simulation results in Section 4.

We thank the Reviewer for mentioning this issue. In our revised paper, we have added the simulation results in Fig.5 for verification.

  1. There are some typos in this paper, e.g., in line 81 of page 2, “queuing queuing scheme” should be “queuing scheme”; in Figure 5, Figures 7-9 and Figure 11, the labels of the longitudinal axis should be unified as Ysum. Please carefully recheck them.

We thank the Reviewer for mentioning this issue. In our revised paper, we have fixed those typos.